# The Potential Role of Urinary Microbiome in Benign Prostate Hyperplasia/Lower Urinary Tract Symptoms

**DOI:** 10.3390/diagnostics12081862

**Published:** 2022-08-01

**Authors:** Seong Hyeon Yu, Seung Il Jung

**Affiliations:** Chonnam National University Medical School, Gwangju 61469, Korea; domer12@hanmail.net

**Keywords:** urinary microbiome, benign prostate hyperplasia, lower urinary tract symptoms

## Abstract

Historically, urine in the urinary tract was considered “sterile” based primarily on culture-dependent methods of bacterial detection. Rapidly developing sequencing methods and analytical techniques have detected bacterial deoxyribonucleic acid and live bacteria in urine, improving our ability to understand the urinary tract microbiome. Recently, many studies have revealed evidence of a microbial presence in human urine in the absence of clinical infections. In women, fascinating evidence associates urinary tract microbiota with lower urinary tract symptoms (LUTS). However, the association between urinary tract microbiota and men with LUTS, particularly those with benign prostate hyperplasia (BPH), has not been established. In addition, the identification of the proinflammatory cytokines and pathogens responsible for the clinical progression of BPH is still underway. This review article aimed to address microbiome-related evidence for BPH. Further studies are required for a comprehensive understanding of the relationship between the urogenital microbiome and BPH pathogenesis to facilitate the development of preventive and therapeutic approaches for male LUTS.

## 1. Introduction

Recent developments in molecular biology techniques and culture methods have allowed for the detection of a specific microbiome associated with body sites previously believed to be sterile, including the urinary tract [1,2,3,4,5]. Based on these developments, many studies have revealed that changes in the microbial population in the urinary tract, known as dysbiosis, are associated with urologic symptoms and disease. In addition, the discovery of bacterial communities and the investigation of their roles in urological diseases have begun to suggest novel factors that may have implications for the pathophysiology and management of these common urological conditions [6,7].

Lower urinary tract symptoms (LUTS) are common in men and benign prostatic hyperplasia (BPH) accounts for a substantial proportion of men with LUTS [6,7]. The prevalence of BPH increases with age and is histologically found in approximately 40% of patients in their 50s, >70% of patients in their 60s, and approximately 90% of patients >80 years [8]. LUTS have generally been attributed to BPH and its secondary effects on bladder function. However, some patients show histological evidence of BPH but do not show LUTS. According to 11 cross-sectional population studies, the incidence of moderate-to-severe LUTS has been reported to be 29% in men in their 50s and 56% in men in their 70s [9]. These reports show that patients with histological BPH are not necessarily consistent with those with LUTS.

Although LUTS and BPH affect the quality of life of numerous patients, [10] the pathophysiologic mechanisms of prostatic diseases are not yet fully elucidated. Chronic prostatic inflammation has been implicated as a significant cause of BPH, but the source of this inflammation has been subject to debate [11,12,13]. Evolving studies indicate that the microbiome can impact prostatic inflammation in relation to BPH [14,15]. In the current review, we summarize the leading recent publications on the urinary microbiome in male BPH.

## 2. Materials and Methods

We performed a nonsystematic narrative review of published articles in English from 1980 to May 2022. The search terms were as follows: (“prostate” OR “benign prostate hyperplasia” OR “chronic prostate inflammation” OR “lower urinary tract symptoms”) AND (“microbiota” OR “urinary microbiota” OR “gut microbiota”). We also searched related articles on PubMed and manually searched the reference list of identified articles to identify additional relevant articles.

### 2.1. The Journey to Discover the Urinary Microbiota of Healthy Individuals

The term “microbiota” refers to the ecological community of symbiotic and pathogenic microorganisms, including bacteria, viruses, archaea, fungi, and protozoa living in our bodies. The term “microbiome” is the genetic material of these microorganisms [16]. The Human Microbiome Project was established in 2008 with the goal of characterizing the human microbiome and analyzing its role in human health and disease [17]. However, the urinary tract was not included in this project, based on the perception that urine is sterile. Historically, bladder urine has been considered “sterile,” based primarily on culture-dependent methods of bacterial detection. The speculation that healthy human urine is sterile arose before the clinical application of culture-independent molecular sequencing, which was likely influenced by standard microbial culture techniques [18].

Recently, numerous studies have revealed evidence of microbiota in the bladders of adults without clinical infection [19,20,21,22,23,24,25,26,27,28,29], and the presence of microbiota in urine has disproved the dogma of urine being sterile. These results can be obtained with 16S ribosomal ribonucleic acid (rRNA) gene sequencing and expanded quantitative urine culture (EQUC), which are techniques primarily utilized to reveal bacterial deoxyribonucleic acid (DNA) and live bacteria in culture-negative urine samples [19]. However, the association between urinary microbiota and clinical factors depends on proper sampling from specific anatomic sites. Although most studies on men are based on voided urine [22,23,24,25,26,27,28,29], voided urine does not truly represent the bladder microbiota, as the bacterial DNA detected in midstream voided urine often differs substantially from the DNA detected in catheterized urine. In fact, a study comparing paired voided and catheterized urine obtained from men found that these paired samples often did not match, providing evidence that voided urine does not truly characterize the male bladder microbiota [30].

As mentioned above, many studies have used midstream urine (MSU) samples or spontaneously voided urine [22,23,24,25,26,27,28,29], in which the first milliliters are discarded, as they may be contaminated with bacteria from the skin. In addition, MSU samples may contain microbial contamination with bacteria from the uroepithelium, periurethral gland, or genital tract. This may lead to the improper characterization of the urinary bladder microbiome due to the involvement of the urogenital microbiota [25,28]. Not even the joint use of funnels (for urine collection) and silver antimicrobial wipes have prevented sample contamination [31]. In men, the risk of contamination in MSU samples is lower than that of women, but there is also a chance of introducing microbial load from nearby tissues, such as the urethra [32,33].

Another widely used method for collecting urine samples involves the use of transurethral catheters. However, this is a more invasive technique, and urethral bacteria may still contaminate the samples during catheter insertion. Therefore, the best option for the collection of urine samples in studies regarding the urinary microbiome is suprapubic aspiration, since it is performed directly from the bladder to avoid contact with other areas. Data regarding the composition of the urinary bladder microbiome obtained using this method are more accurate than those obtained through other routines [34]. However, it is the most invasive, painful, and complicated collection method. A comparative study of microbial communities in urine obtained either with suprapubic aspiration or transurethral catheter showed that the outcome is very similar, regardless of the collection method, which makes transurethral catheterization the most appropriate option for urine collection. This method is widely accepted for the study of the urinary bladder microbiome [32].

### 2.2. Urinary Microbiota in Healthy Male Individuals

To date, numerous studies have reported on the urinary microbiota of healthy men and women [19,20,21]. However, there are significantly fewer published studies on the male urinary microbiota than those on females. General information on the urinary microbiota of healthy male individuals is presented in Table 1. Nelson and Dong et al. reported the first results on the male urinary microbiome using first-void urine samples from patients with and without sexually transmitted infections [22,24]. The characterization of the male urinary microbiota in healthy individuals, as well as dysbiosis associated with disease, has also been enabled by 16S rRNA gene sequencing.

Although most microbiomes in the urinary tracts of healthy men and women are similar, slight differences in the compositions of the urinary tract microbiomes between these populations have been found. Common female urinary microbiomes, such as *Prevotella, Escherichia, Enterococcus, Streptococcus,* or *Citrobacter* have also been found in men. However, while male microbiota are characterized by the predominance of *Corynebacterium* and *Streptococcus*, *Lactobacillus* is the most abundant microbiome in the female urinary tract [25,28]. *Lactobacillus* and *Pseudomonas* have also been found in male urinary microbiota, but their proportions are lower than those of women [28]. Other shared microbiomes in the male and female urinary tract are coagulase-negative *Staphylococci* and *Eubacterium* [35].

These differences can be attributed to anatomical factors and urine sampling methods. In females, the urinary tract is anatomically close to the vagina; therefore, the vagina is suggested to be the main source of the female urinary microbiome. Thomas-White et al. analyzed cultured bacteria from the female bladder and compared them to gastrointestinal and vaginal microbiota, demonstrating a close similarity between vaginal and bladder microbiota, with functional capacities that are distinct from those observed in gastrointestinal microbiota [36]. However, another study reported evidence that the origin of urinary microbiota is the gut [37]. Furthermore, gut microbiota composition and its metabolites can be related to the urinary microbiome and various genitourinary pathologies [38]. Therefore, further studies are needed to determine the origin of the urinary microbiota and to explore the interaction between the microbiota of urine and other sites in healthy humans.

Another hypothesis is the difference in urine sampling methods. Contrary to studies on men, numerous studies on female urinary microbiota have used urine samples obtained by catheterization. As mentioned above, voided urine does not truly represent the urinary microbiota. In addition, MSU samples may contain microbial contamination with bacteria from the uroepithelium, periurethral gland, or genital tract, thus misleading the proper characterization of the urinary bladder microbiome [25,28]. In a recent study, Bajic et al. reported the importance of catheterized urine samples in both male and female urinary microbiota using EQUC and 16S rRNA sequencing. However, the small sample size of their study is a limitation for the true characterization of male urinary microbiota [30]. Therefore, a standardized method for urine sample collection needs to be developed.

### 2.3. The Urinary Microbiota and LUTS in Male Individuals

The new perception that “urine is not sterile,” as well as new knowledge associated with the urinary microbiome, has led to a better understanding of LUTS. Compelling evidence associates lower urinary tract microbiota with LUTS in women [19]. Several studies have reported that changes in the microbiota in urine (dysbiosis or less diversity) are related to urge urinary incontinence (UUI) and interstitial cystitis/bladder pain syndrome (IC/BPS) [39,40,41]. In addition, the invasion of the bladder wall by uropathogenic *Escherichia coli* (*E. coli*) (UPEC) and the formation of biofilm-like intracellular bacterial communities (IBCs) by occult and recurrent cystitis may be possible explanations for LUTS (e.g., overactive bladder (OAB) and IC/BPS). IBCs after urinary tract infection (UTI) go undetected in urine cultures and evade antibiotic therapy; therefore, they could be a source of recurrent UTI and a possible aggravation factor in LUTS [42,43,44]. However, a similar relationship between LUTS and urinary microbiota in men has not yet been established. Previous information regarding urinary microbiota in men with LUTS is weak, which makes interpretation in clinical practice difficult. Therefore, further investigation on the relationship between male urinary microbiota and LUTS should be conducted for a possible clinical application to LUTS in men, as in studies on women.

BPH and its secondary effects on bladder function have been considered to be causes of LUTS. However, the contribution of BPH severity to LUTS is not equivocal, and LUTS are complex processes involving multiple factors, such as prostate inflammation and fibrosis [45]. Therefore, instead of placing emphasis on BPH, LUTS should be focused on as a multifactorial etiology with associated symptoms. Among these multifactorial etiologies, one possible etiology is that the inhabitant bacterial microbiota in the lower urinary tract impacts the local immunological environment [13]. The results of studies on urinary microbiota in men with LUTS/BPH are presented in Table 2.

16S rRNA, 16S ribosomal RNA; BPH, benign prostatic hyperplasia; DNA, deoxyribonucleic acid; EPS, EQUC, expanded quantitative urine culture; IPSS, International Prostate Symptom Score; LUTS, lower urinary tract symptoms; MSU, midstream urine; PC, prostate cancer; TUC, catheterized urine.

Lewis et al. examined the effect of aging on the microbiota of male midstream voided urine and found that microbiota in the male urethra appear to diminish in number and increase in diversity with age [26]. This finding suggests that changes in the urethra and bladder microbiota that come with age might be associated with increasing LUTS in older men, which is typically due to BPH. However, no study has investigated the effect of aging on the microbiota of catheterized male urine. According to studies on females, urinary microbiota may play a role in the pathogenesis of OAB, although the mechanisms underlying the causative relationship, as well as its possible therapeutic implications, are still uncertain [19]. Decreased urinary microbiome diversity due to catheterized urine is frequently detected in patients with UUI [39,41]. Therefore, urinary microbiome diversity in the male urinary system should be evaluated to find its clinical significance in male LUTS.

Using a combination of 16S rRNA sequencing and EQUC, Bajic et al. confirmed the presence of dissimilar microbiota in the urethra and provided evidence of microbes in the bladders of males [30]. Using the International Prostate Symptom Score (IPSS), 28 men undergoing surgery for benign prostatic enlargement (BPE)/LUTS and 21 undergoing non-BPE/LUTS surgery were stratified, and paired voided/catheterized urine specimens were collected for EQUC and 16S ribosomal RNA gene sequencing. The presence of distinct urinary tract microbiota that differ between BPH/LUTS patients based on the degree of symptoms suggests the presence of a link between urinary microbiota and male LUTS. Bajic et al. showed that the severity of LUTS (measured by IPSS) is associated with the presence of bacteria in catheterized urine but not in midstream voided urine. Moreover, patients in the increasing IPSS category were more likely to have detectable bacteria (2.2-fold higher odds) [30]. This study was the first to demonstrate an association between the male urinary microbiome and LUTS and is important in establishing the difference between voided and catheterized urine collection in men. MSU, catheterized urine, seminal fluid, expressed prostatic secretion, stool, and resected prostatic tissue have been used to profile the microbiome in prostatic diseases. However, Bajic et al. indicated that voided urine cannot adequately characterize the male bladder microbiome due to the existence of distinct microbiota in the anterior urethra of the lower male urinary tract. In addition, they suggested that catheterized urine may be the most appropriate way to sample male bladder microbiota because voided urine shows significant contamination compared to catheterized urine.

Holland et al. examined the correlation between urinary and fecal microbial profiles and the various aspects of lower urinary tract symptoms (LUTS) in men [47]. Among the 48 fecal OTUs, which showed a significant correlation with one or more of the IPSS, the most substantial negative correlation was between *Lachnospiraceae blautia*—a bacterium that increases the availability of gut anxiolytic and antidepressant short-chain fatty acids—and symptom severity. The abundance of *L. blautia* continued to have a protective correlation with LUTS when examining different levels of IPSS severity. However, voided urine samples were used, and the results should therefore be interpreted with caution as microbial contamination may have occurred. Recently, Lee et al. showed that some bacterial genera were present in MSU samples of the BPH group [49]. Some bacterial genera correlated with a high IPSS, as well as severe storage and voiding symptoms. However, voided MSU cannot adequately characterize the male bladder microbiome, and evidence of an association between urinary microbiota and symptom severity (e.g., IPSS) is still lacking. Furthermore, there has been no study including used naïve vs. treated patients (alpha-blocker and other agents used in male LUTS), and neither is there evidence of clinical response vs. clinical progression according to urinary microbiota after medicating male LUTS. Therefore, further research targeting specific microbes to identify their role in the development and treatment of LUTS is necessary.

### 2.4. The Urinary Microbiota and BPH in Male Individuals

BPH is the most common urological condition affecting aging males. In addition, LUTS due to BPE increase with age. LUTS have also been associated with bladder outlet obstruction (BOO), often caused by BPE resulting from histological BPH [50,51]. Combination therapy with α-blockers and 5-α reductase inhibitors significantly reduces the risk of clinical progression; BPH-related complications, such as hematuria, bladder stones and upper tract deterioration; and BPH-related surgery [52]. Nevertheless, in a previous study, 13% of men on combination therapy showed clinical progression after 4 years of treatment, with 5% requiring surgical intervention [53]. As such, the lack of response to medical therapy, as well as the disease progression despite medical therapy, suggests that its primary pathophysiology is heterogeneous, diverse, and poorly understood [54]. Our current understanding of the pathophysiology of BPE-associated LUTS relies on the fixed belief that the bladder is sterile. Recently, this belief was invalidated by several reports describing the presence of microbiota in urine acquired from male bladders in the absence of UTI.

Historically, prostatic inflammation has been considered to play a significant role in BPH pathogenesis and progression [11,12,55]. The association between histological prostatitis and BPH has received considerable attention due to the high frequency of inflammatory histological findings in prostatic biopsy specimens from BPH patients [56]. Additionally, in the majority of patients, BPH/LUTS are known to be related to intraprostatic infiltration of inflammatory cells [57]. In the MTOPS study, 1197 patients were biopsied, of whom 544 showed histological evidence of prostatic inflammation. Patients with prostatic inflammation are significantly more likely to develop acute urinary retention than those without prostatic inflammation [57]. Although there is evidence of a connection between inflammation and BPH, the etiology of inflammation remains poorly understood. One possible etiology is that the presence of an inflammatory status can stimulate IL-6 and IL-8 production. Interleukins can stimulate androgen receptors independently via dihydrotestosterone (DHT) and transforming growth factor (TGF) synthesis. Therefore, inflammation can conserve tissue proliferation in the presence of 5-α reductase inhibitors. Moreover, proliferation and hypoxia can stimulate the production of reactive oxygen species (ROS), further supporting a vicious cycle [11]. Another possible etiology is that the gut microbiota alteration, induced by several factors such as age, diet, lifestyle, and drug intake, can influence prostate inflammation status indirectly by activating the immune system [38]. Gut microbiota alterations could promote an inflammatory condition in distant sites, including the prostate, through the production of proinflammatory cytokines, such as IL-17, IL-23, TNF-alpha, and IFN-gamma [58]. In addition, Takezawa et al. reported that patients with BPH had a high Firmicutes/Bacteroidetes ratio in gut microbiota, and the short-chain fatty acids from gut bacteria promoted prostate enlargement by activating the IGF-1 signaling pathway. These findings can be considered to be evidence of the existence of a “gut–prostate axis” [59].

This inflammatory process in BPH is a potential new target for diagnosis and treatment. A recent report hypothesized an association between the prostate and proinflammatory bacterial species [13], including several bacterial species previously related to female UUI [60]. Recently, some studies in men have been performed to explore the role of urinary microbiota in BPH.

Yu et al. utilized 16S rRNA sequencing to compare the microbes in EPS, seminal fluid, and voided urine in BPH and prostate cancer patients. They reported that the bacterial flora in the EPS of patients with BPH differed from those of patients with prostate cancer. BPH patients had higher rates of the genera *Eubacterium* and *Defluviicoccus* but lower rates of the phyla Bacteroidetes, *Alphaproteobacteria*, and Firmicutes, as well as the family *Lachnospiraceae* and the genera *Propionicimonas*, *Sphingomonas*, and *Ochrobactrum* [46]. Certain bacteria can induce a chronic inflammatory state in the prostate, resulting in a higher production of proinflammatory cytokines. These results suggest that ecological dysbiosis of the microbiota in the prostatic fluid might play a key role in the pathophysiology of BPH and prostate cancer.

In another study, Jain et al. showed that BPH tissues have divergent microbial compositions, including the commonly found E. coli (phylum *Proteobacteria*), which might contribute to BPH-associated inflammation and/or tissue damage [48]. Microbial cultures of tissue samples showed the presence of live bacteria in 55.5% of the patient tissues. The majority of isolates were coagulase-positive Staphylococcus, E. coli, and Micrococcus species. The presence of multiple bacteria has been found in the BPH tissues, with the most common phyla being Proteobacteria, Actinobacteria, Firmicutes, and Bacteroidetes [48]. Phospho-histone γH2A.X staining confirmed the presence of cells with damaged DNA lesions in BPH tissues and correlated it with inflammation severity. Furthermore, the BPH-associated *E. coli* induced in vitro NF-κB signaling and DNA damage in prostate epithelial cells [48].

The presence of pathogenic bacteria may contribute to the pathogenesis of BPH through acute and chronic inflammation [61]. However, the repeated antibiotic therapies that often accompany the clinical course of urogenital and prostate infections may also lead to frequent dysbiosis [62]. Recently, several therapeutic approaches, such as the supplementation of live probiotics, bacteria, and hydrocolon therapy, were theoretically suggested to restore eubiosis and stop the vicious circle of dysbiosis–urogenital infections [62].

Although the prostate is an androgen-dependent organ, it frequently shows proliferative changes (in cases of BPH) in an environment where testicular function and blood androgen levels decrease with age. This androgen-independent association between prostate proliferation and inflammation has received much attention in the literature [63]. The imbalance of sex hormones, especially the involvement of estrogen—which is relatively increased due to a decrease in androgen concentration with age—is suggested as a cause of the acute or chronic inflammation that occurs frequently in the prostate, in addition to external factors such as infection and trauma [63,64,65]. Therefore, further research regarding the relationship between prostate proliferation and inflammation, as well as the composition and changes in the urinary microbiota with various external and internal factors, is required.

### 2.5. The Role of the Inflammasome in LUTS/BPH

As previously mentioned, histological BPH is not necessarily consistent with male LUTS, and an increasing number of studies have shown that LUTS are often unconnected to the prostate [66]. Bladder dysfunction may also cause LUTS, including detrusor overactivity/OAB and detrusor underactivity/underactive bladder [66]. Storage and voiding symptoms occur in common conditions, such as UTIs and BOO, and these conditions are confirmed to have underlying inflammations that directly trigger these symptoms. While it is well understood that inflammation causes bladder dysfunction in several common diseases, the role of inflammasomes as central mediators has only recently been explored.

Inflammasomes may play a role in the inflammation caused by bladder and prostate microbiota. By understanding the role of each inflammasome in various pathological conditions, we may be able to target them using therapeutic agents to prevent these diseases and their symptoms. According to a recent study, inflammatory processes mediated by inflammasomes can lead to storage/voiding symptoms, bladder fibrosis, and denervation [67]. Inflammasomes are multiprotein oligomers responsible for initiating inflammatory responses. Additionally, inflammasomes promote the maturation and secretion of the proinflammatory cytokines interleukin 1β (IL-1β) and interleukin 18 (IL-18), as well as pyroptosis, which is programmed proinflammatory cell death distinct from apoptosis [67]. The inflammasomes formed by the nucleotide-binding domain, leucine-rich-containing family, and pyrin domain containing-3 (NLRP3) are known as NLRP3 inflammasomes. NLRP3 inflammasomes are located intracellularly, where they recognize two different types of signals: pathogen-associated molecular patterns (PAMPs) and danger (or damage)-associated molecular patterns (DAMPs) [68]. PAMPs include common bacterial components and known virulence factors such as lipopolysaccharide (LPS), flagellin, and hemolysins [69]. DAMPs include adenosine triphosphate (ATP), uric acid crystals, high-mobility group box 1, and heat-shock proteins, and are typically released by stressed, damaged, or dying cells [68]. An NLRP3 inflammasome contains three components: NLRP3 (node-like receptor), ASC, and procaspase-1. It requires an adaptor protein to connect to procaspase-1. The adapter is an apoptosis-associated speck-like protein containing CARD (ASC). Activated caspase-1 cleaves pro-IL-1β and pro-IL-18 to their active forms (IL-1β and IL-18). Caspase-1 is also responsible for the formation of pores in the cell membrane, which results in pyroptosis [67].

Inflammasome-activating processes induced by BOO are DAMPs induced by hypoxia/reperfusion, increased pressure, and repetitive stretching. BOO triggers an ongoing inflammatory process mediated by NLRP3, which evokes negative downstream events, such as storage/voiding symptoms and fibrosis. Although the most common cause of BOO is BPH, BPH itself may be a result of inflammasome activation [70]. Infectious prostatitis can activate inflammasomes through PAMPs. Inflammasomes may mediate both infectious and sterile prostatitis, leading to BPH, BOO, and activation of NLRP3 within the bladder, eventually causing harmful effects, such as storage/voiding symptoms, bladder fibrosis, and denervation. However, the main NLR involved in the prostate appears to be NLRP1 and not NLRP3. NLRP1 was the first inflammasome to be described, and it plays a role in prostate inflammation [71,72]. NLRP1 differs mechanistically from NLRP3, as it needs to undergo autolytic proteolysis for activation, and it may play a role in mediating UTI responses [73].

Recently, intracellular bacteria have provided explanations for OAB symptoms. For example, α-hemolysin is expressed in UPEC and activates NLRP3, which triggers pyroptosis in human urothelial cells as a host defense mechanism. In addition, UPEC releases multiple mediators from the urothelium and promotes urothelial barrier defects. Immune cell infiltration and the release of proinflammatory cytokines (e.g., nerve growth factor (NGF)) are known to sensitize peripheral afferents and, in this way, enhance bladder sensation [74]. After the exfoliation of urothelial cells, quiescent cellular reservoirs can develop, leading to chronic cystitis [75]. Intracellular bacteria and quiescent cellular reservoirs may cause chronic LUTS due to persistent occult infection, or perhaps due to chronic irritation and the inflammation of the bladder wall from the initial result of infection and invasion. In a small sample of patients with OAB, Khasriya et al. found intracellular bacteria in 94% of patients, compared with 29% in controls [76]. Considering this, intracellular bacteria seem to be found more commonly in patients with OAB than in controls.

However, the role of intracellular bacteria in causing LUTS is poorly understood. Contreras-Sanz et al. found that LUTS are caused by the altered urothelial ATP signaling pathway due to intracellular bacteria [77]. Additionally, they found that ATP mediates the sensations of bladder filling and urgency, and a low-grade inflammatory response (pyuria ≥ 10 wbc/uL) in 10–35% of MSU specimens from patients with OAB showed the worst symptoms of frequency. In addition, intracellular bacteria were visualized in shed urothelial cells from approximately 80% of OAB patients with pyuria. The basal release of ATP was significantly greater in the urothelium of OAB patients with pyuria than in non-OAB patients or OAB patients without pyuria. Therefore, they suggested that the increased basal release of ATP in the urothelium is responsible for intracellular bacteria in OAB patients with pyuria [77].

### 2.6. Prospectives of Urinary Microbiota

The term “microbiota” includes not only bacteria but also human and bacteriophage viruses, and even fungi. However, most human studies have focused exclusively on bacteria, and few investigations of fungi and viruses have been conducted [78,79,80]. Bacteriophage viruses play an important role in the stability of microbial communities [81]. In addition, phages can destroy UPEC biofilms, and the use of phages may be the best option for the prevention and treatment of LUTS caused by UTI without significant adverse effects [82,83]. Regarding human viruses, Moustafa et al. examined whole metagenome shotgun sequencing from 49 MSU samples and identified different human viruses such as human papillomavirus, BK polyomavirus, and JC polyomavirus, related to papilloma, UTI, and OAB, respectively [84,85,86]. Studies of urinary fungal microbiota are scarcer than those of microbiota. Therefore, the discovery of microbiomes other than bacteria in urine could open new areas of research for the analysis of new mechanisms in urologic symptoms and disease, as well as for innovative preventive measures and new therapeutic strategies.

## 3. Conclusions

The urinary tract has been found to be unsterile. The potential influence of the microbiome in the urinary tract on the pathobiology of genitourinary diseases is only now beginning to be appreciated. Preliminary evidence has shown that the microbiome plays a role in various aspects of male urological diseases. However, to date, the evidence is weak, making translation into clinical practice difficult. The materials used for microbiota profiling may have affected the findings, providing limitations for these studies. Standardized methodologies, such as sample collection and the sequencing of the same variable region, as well as reporting methods, should be developed. In addition, most studies have focused exclusively on bacteria, and the contribution of other microbial populations, such as fungi and viruses, to the urinary microbiome should be elucidated. Further investigations on urogenital microbiota may reveal potential associations between BPH, male LUTS, and causative pathogens. Perhaps, with further studies, a means to handle the urinary microbiome to improve patient outcomes can be developed.

## Figures and Tables

**Table 1 diagnostics-12-01862-t001:** Information regarding the urinary microbiota in healthy males [19,20,21].

Subject	Study Population	Main Bacterial Taxa	Sample Collection	Technique Used
Nelson et al. (2010) [22]	Men aged ≥ 18 y without STI (*n* = 9)	*Corynebacterium*, *Lactobacillus*, *Streptococcus*, *Staphylococcus*, *Propionibacterium*	First-void urine	16S rRNA GS
Nelson et al. (2012) [23]	Healthy adolescent men (aged 14–17 y) (*n* = 18)	*Lactobacillus*, *Streptococcus*, *Sneathia*, *Mycoplasma*, *Ureaplasma*	First-void urine	16S rRNA GS
Dong et al. (2011) [24]	Men without STI (*n* = 10)	*Lactobacillus*, *Sneathia*, *Veillonella*, *Corynebacterium*, *Prevotella*, *Streptococcus*, *Ureaplasma*, *Mycoplasma*, *Anaerococcus*, *Atopobium*, *Aerococcus*, *Staphylococcus*, *Gemella*, *Enterococcus*, *Finegoldia*	First-void urine	16S rRNA GS
Fouts et al. (2012) [25]	Healthy men aged 24–50 y (*n* = 11)	*Lactobacillus*, *Corynebacterium*, *Staphylococcus*	MSU	16S rRNA GS
Lewiset al. (2013) [26]	Healthy men (*n* = 6)	Forty-six genera identified	MSU	16S rRNA GS
Gottschicket al. (2017) [27]	Healthy men (*n* = 31)	*Prevotella amnii*, *Sneathia* *amnii*, *Shigella sonnei*, *Enterococcus* *faecalis*, *Streptococcus agalacticie*, *Citrobacter murliniae*	MSU	16S rRNA GS
Modenaet al. (2017) [28]	Healthy men (*n* = 10)	*Streptococcus*, *Lactobacillus*, *Prevotella*, *Corynebacterium*, *Pseudomonas*	MSU	16S rRNA GS
Frølund et al. (2018) [29]	Healthy men (*n* = 46)	*Gardnerella*, *Lactobacillus*, *Sneathia*, *Finegoldia*, *Alphaproteobacteria*, *Prevotella*, *Enterococcus*	First-void urine	16S rRNA GS

**Table 2 diagnostics-12-01862-t002:** Studies investigating the microbiota of males with benign prostatic hyperplasia/lower urinary tract symptoms [14,15].

Subject	Sample Size (n)	Sample Type	Analysis Method	Relevant Microbiota	Primary Finding
Lewis et al. (2013) [26]	6	MSU	16S rRNA gene sequencing	Firmicutes, Proteobacteria, Actinobacteria, Fusobacteria, Bacteroidetes	Diminish in numbers and increase in diversity with age
Bajic et al. (2020) [30]	49	MSU, TUC	EQUC, 16S rRNA gene sequencing	*Streptococcus*, *Veillonella*, *Gardnerella*, *Staphylococcus*, *Candida*	An increase in IPSS was associated with significantly high odds of detectable bacteria in TUC TUC is adequate to sample the bladder microbiome
Yu et al. (2015) [46]	21 BPH, 13 Pc	Voided urine, EPS/seminal fluid	16S rRNA gene sequencing with PCR-DGGE analysis	*Eubacterium*, *Defluviicoccus*	Bacterial flora in the EPS of patients with BPH differ from those with PC
Holland et al. (2019) [47]	30 men	Urine & fecalsamples	16S rRNA gene sequencing	*Lachnospiraceae*, *Bacteoidaceae*, *Erysipelotrichaceae*, *Ruminococcaeceae*, *Prevotellaceae*, *Clostridiales*, *Veillonellaceae*, *Enterococcaceae*, *Corynebacteriaceae*, *Incertae*	The abundance of *L. blautia* continued to have a protective correlation with LUTS when looking at various levels of IPSS severity
Jain et al. (2020) [48]	36 men	Resected tissue	EQUC, 16S rRNA gene sequencing, immunohistological staining	Coagulase-positive *Staphylococcus*, *E. coli*, Micrococcus species, Proteobacteria, Actinobacteria, Firmicutes, Bacteroidetes	The presence of live bacteria in 55.5% of patient tissues Staining confirmed the presence of cells with damaged DNA lesions in BPH tissues and correlated with the severity of inflammation
Lee et al. (2021) [49]	77 men with BPH, 30controls	MSU	16S MetagenomicSequencing	*Haemophilus*, *Staphylococcus*, *Dolosigranulum*, *Listeria*, *Phascolarctobacterium*, *Enhydrobacter*, *Bacillus*, *[Ruminococcus] torques*, *Faecalibacterium*, *Finegoldia*	Some bacterial genera correlated with a high IPSS as well as severe storage and voiding symptoms

## Data Availability

Not applicable.

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
