# Peer review of "The Potential Role of Urinary Microbiome in Benign Prostate Hyperplasia/Lower Urinary Tract Symptoms"

_diagnostics, 2022, doi:10.3390/diagnostics12081862_

Round 1
Reviewer 1 Report
This is an interesting topics but authors need to explain how they did the search, which key words were used to search the literature and who made the revision and final selection of the papers.
How many papers were found and how many were excluded? And why were they excluded. Please present these data in a diagram
Authors need to clarify if they want to do a metanalysis or simply a narrative review. In the latter case explain how the organization was done.
Authors need to indicate how many papers included used naïve vs treated patients (alpha-blocker and other agents used in male LUTS), is the IPSS different accoding to microbioma?, is there an evidence of clinical clinical response versus clinical progression according to microbioma? These points are not clearly addressed.
Also there is no clear description of microbioma according to te degree of bladder outlet obstruction. Is there any study?
Terminology is important. BPH is a microscopic disease. Prostate enlargement and obstruction is different. More, there are no irritative symptoms but rather storage and voiding LUTS.
Author Response
All authors sincerely thank you for your kind comments of the manuscript
Reviewer: 1
Comments to the Author
This is an interesting topics but authors need to explain how they did the search, which key words were used to search the literature and who made the revision and final selection of the papers. How many papers were found and how many were excluded? And why were they excluded. Please present these data in a diagram. Authors need to clarify if they want to do a metanalysis or simply a narrative review. In the latter case explain how the organization was done.
A) This paper is a simple narrative review. As your recommendation, we have revised and inserted the following sentences in the manuscript (Materials and Methods)
[Materials and Methods]
We performed a non-systematic narrative review of published articles in English from 1980 to May 2022. The search terms were as follows; (“prostate” OR “benign prostate hyperplasia” OR “chronic prostate inflammation” OR “lower urinary tract symptoms”) AND (“microbiota” OR “urinary microbiota” OR “gut microbiota”). We also searched related articles on PubMed, and manually searched the reference list of identified articles to identify additional relevant articles.
Authors need to indicate how many papers included used naïve vs treated patients (alpha-blocker and other agents used in male LUTS), is the IPSS different accoding to microbioma?, is there an evidence of clinical clinical response versus clinical progression according to microbioma? These points are not clearly addressed.
A) Thank you for your comment. However, unfortunately, there has been no study including used naïve vs treated patients (alpha-blocker and other agents used in male LUTS). Furthermore, there has been no evidence of clinical response versus clinical progression according to urinary microbiome. Instead, we have mentioned the study about presence of distinct urinary tract microbiota that differ between BPH/LUTS patients based on the degree of symptoms (measured by IPSS) in our manuscript as follows.
[The urinary microbiota and LUTS in male individuals]
Using a combination of 16S rRNA sequencing and EQUC, Bajic et al. confirmed the presence of dissimilar microbiota in the urethra and provided evidence of microbes in the bladders of males. [30] Using the International Prostate Symptom Score (IPSS), 28 men undergoing surgery for benign prostatic enlargement (BPE)/LUTS and 21 undergoing non-BPE/LUTS surgery were stratified, and paired voided/catheterized urine specimens were collected for EQUC and 16S ribosomal RNA gene sequencing. The presence of distinct urinary tract microbiota that differ between BPH/LUTS patients based on the degree of symptoms suggests the presence of a link between the urinary microbiota and male LUTS. Bajic et al. showed that the severity of LUTS (measured by IPSS) is associated with the presence of bacteria in catheterized urine but not in midstream voided urine. Moreover, patients in the increasing IPSS category were more likely to have detectable bacteria (2.2-fold higher odds). [30]. This study is the first to demonstrate an association between the male urinary microbiome and LUTS, and is important in establishing the difference between voided and catheterized urine collection in men.
Also, there is no clear description of microbioma according to te degree of bladder outlet obstruction. Is there any study?
A) Thank you for your comment. However, to our knowledge, there has been no study for clear description of microbiome according to the degree of bladder outlet obstruction.
Terminology is important. BPH is a microscopic disease. Prostate enlargement and obstruction is different. More, there are no irritative symptoms but rather storage and voiding LUTS.
A) Thank you for your comment. As your recommendation, we revised the word “irritative” to “storage” or “storage/voiding” in entire manuscript.
We also upload the final revised manuscript as a file.

Reviewer 2 Report
The role of the urobiome is increasingly recognized in relation to prostate cancer, but much less studies in benign conditions such as BPH or prostatitis. Therefore, the submitted manuscrit is important, clinically useful and within the scope of the journal. The authors provide a lot of information and that is why the paper finally deserves publication. Parts of the work should be rewritten, however, since the urinary microbiome (in the present form of the manuscript) seems to be limited to the bladder or a topographically unspecified urinary microbiome. As we know from research on the microbiome of the female genital tract (so far mainly focused on the vaginal microbiome), an orchestration of the microbiome at all levels of the genital tract is essential. The interplay betwen the penile microbiome (along with modyfying factors), uretheral, baldder and prostatic microbiome should be mentioned. Further concerns: a) The hormonal influence on the composition of the urine microbiome in men is not discussed. b) The interplay between the gut and urinary microbiomes in health and disease is not addressed, although the gut microbiome deserves a source, or at least a pathway, for the urogenital microbiota in both males and females. c) The role of sexual encounters in changes in the urobiome is not addressed. d) Finally, the possibilities of influencing the urinary and prostatic microbiota (role of sexual behavior, microbiota supplementation, useful and useless treatments, hormonal substitution, etc.) are not sufficiently addressed. The work does not include several directly related publications, so I think the paper would benefit from reflecting the following articles: 1) Russo et al. The relationship between the gut microbiota, benign prostatic hyperplasia, and erectile dysfunction. Int J Impot Res. 2022 Apr 13. PMID: 35418604. 2) Li et al. Alterations of gut microbiota diversity, composition and metabonomics in testosterone-induced benign prostatic hyperplasia rats. Mil Med Res. 2022 Mar 28;9(1):12. PMID: 35346378 3) Gu et al. High-Fat Diet Induced Gut Microbiota Alterations Associating With Ghrelin/Jak2/Stat3 Up-Regulation to Promote Benign Prostatic Hyperplasia Development. Front Cell Dev Biol. 2021 Jun 24;9:615928. PMID: 34249898 4) Magri et al. Multidisciplinary approach to prostatitis. Arch Ital Urol Androl. 2019 Jan 18;90(4):227-248. PMID: 30655633. 5) Hurst et al. Microbiomes of Urine and the Prostate Are Linked to Human Prostate Cancer Risk Groups. Eur Urol Oncol. 2022 Apr 18:S2588-9311(22)00056-6. PMID: 35450835. 6) Terrisse et al. Immune system and intestinal microbiota determine efficacy of androgen deprivation therapy against prostate cancer. J Immunother Cancer. 2022 Mar;10(3):e004191. PMID: 35296557 7) Porter CM et al. The microbiome in prostate inflammation and prostate cancer. Prostate Cancer Prostatic Dis. 2018 Sep;21(3):345-354. PMID: 29795140. 8) Shoemaker and Kim. Urobiome: An outlook on the metagenome of urological diseases. Investig Clin Urol. 2021 Nov;62(6):611-622. PMID: 34729961Author Response
All authors sincerely thank you for your kind comments of the manuscript
Reviewer: 2
Comments to the Author
The role of the urobiome is increasingly recognized in relation to prostate cancer, but much less studies in benign conditions such as BPH or prostatitis. Therefore, the submitted manuscrit is important, clinically useful and within the scope of the journal.
A) Thank you for the great comment.
The authors provide a lot of information and that is why the paper finally deserves publication. Parts of the work should be rewritten, however, since the urinary microbiome (in the present form of the manuscript) seems to be limited to the bladder or a topographically unspecified urinary microbiome. As we know from research on the microbiome of the female genital tract (so far mainly focused on the vaginal microbiome), an orchestration of the microbiome at all levels of the genital tract is essential. The interplay betwen the penile microbiome (along with modyfying factors), uretheral, baldder and prostatic microbiome should be mentioned.
A) This review is “The potential role of urinary microbiome in benign prostate hyperplasia/lower urinary tract symptoms”. Therefore, this review is mainly related to the urinary microbiome, 16S rRNA gene sequencing and EQUC using urine is the most suitable and universal method to explore the urinary microbiome. However, since the urine is collected from the bladder, as your comment, it may seem to be limited to the bladder or a topographically unspecified urinary microbiome. However, to our knowledge, there has been no study for the interplay between the penile microbiome (along with modifying factors), urethral, bladder and prostatic microbiome.
Further concerns: a) The hormonal influence on the composition of the urine microbiome in men is not discussed.
A) Thank you for your comment. However, to our knowledge, there has been no study for the hormonal influence on the composition of the urine microbiome in men.
b) The interplay between the gut and urinary microbiomes in health and disease is not addressed, although the gut microbiome deserves a source, or at least a pathway, for the urogenital microbiota in both males and females.
A) We already have mentioned the correlation between urinary and fecal microbial profiles and the various aspects of lower urinary tract symptoms (LUTS) in men (Reference 46) in the “The urinary microbiota and LUTS in male individuals” section. However, as your comment, we have revised and inserted the following sentences in the manuscript (Urinary microbiota in healthy male individuals, The urinary microbiota and BPH in male individuals)
[Urinary microbiota in healthy male individuals]
However, another study reported evidence that the origin of the urinary microbiota is the gut. [37] Furthermore, gut microbiota composition and its metabolites can be related to urinary microbiome and various genitourinary pathologies. [38] Therefore, further studies are needed to determine the origin of the urinary microbiota and to explore the interaction between the microbiota of urine and other sites in healthy humans.
[The urinary microbiota and BPH in male individuals]
Another possible etiology is that the gut microbiota alteration, induced by several factors such as age, diet, lifestyle, and drug intake, can influence prostate inflammation status indirectly by the activation of immune system. [38] Gut microbiota alterations could promote an inflammatory condition in distant sites including prostate, through the production of proinflammatory cytokines, such as IL-17, IL-23, TNF-alpha, and IFN-gamma. [56] In addition, Takezawa et al. reported that the patients with BPH had high Firmicutes/Bacteroidetes ratio in gut microbiota, and the short-chain fatty acids from gut bacteria promoted prostate enlargement by activating the IGF-1 signaling pathway. These findings can be considered as evidence of the existence of “gut-prostate axis”. [57]
c) The role of sexual encounters in changes in the urobiome is not addressed.
A) We already have mentioned the urinary microbiome of healthy men and women in “Urinary microbiota in healthy male individuals” section as follows.
[Urinary microbiota in healthy male individuals]
Although most microbiomes in the urinary tracts of healthy men and women are similar, slight differences in the compositions of the urinary tract microbiomes between these populations have been found. Common female urinary microbiomes, such as Prevotella, Escherichia, Enterococcus, Streptococcus, or Citrobacter have also been found in men. However, while the male microbiota is characterized by the predominance of Corynebacterium and Streptococcus, Lactobacillus is the most abundant microbiome in the female urinary tract. [25,28] Lactobacillus and Pseudomonas have also been found in the male urinary microbiota, but their proportions are lower than that in women. [28] Other shared microbiomes in the male and female urinary tract are coagulase-negative staphylococci and Eubacterium. [35]
d) Finally, the possibilities of influencing the urinary and prostatic microbiota (role of sexual behavior, microbiota supplementation, useful and useless treatments, hormonal substitution, etc.) are not sufficiently addressed.
A) Thank you for your comment. As your comment, we have revised and inserted the following sentences in the manuscript (The urinary microbiota and BPH in male individuals).
[The urinary microbiota and BPH in male individuals]
The presence of pathogenic bacteria may contribute to the pathogenesis of BPH through acute and chronic inflammation. However, repeated antibiotic therapies that often accompany the clinical course of urogenital and prostate infections may also be led to frequent dysbiosis. [61] Recently, several therapeutic approaches such as supplementation of live probiotics bacteria and hydrocolon therapy were theoretically suggested to restore the eubiosis and stop the vicious circle of dysbiosis-urogenital infections. [61]
Although the prostate is an androgen-dependent organ, it frequently shows proliferative changes (in cases of BPH) in an environment where testicular function and blood androgen levels decrease with age. This androgen-independent association between prostate proliferation and inflammation has received much attention in the literature. [62] The imbalance of sex hormones, especially involvement of estrogen—which is relatively increased due to a decrease in androgen concentration with age—is suggested as a cause of the acute or chronic inflammation which occurs frequently in the prostate, in addition to external factors such as infection and trauma. [62-64] Therefore, further research regarding the relationship between prostate proliferation and inflammation, as well as the composition and changes in the urinary microbiota with various external and internal factors, is required.
The work does not include several directly related publications, so I think the paper would benefit from reflecting the following articles:
1) Russo et al. The relationship between the gut microbiota, benign prostatic hyperplasia, and erectile dysfunction. Int J Impot Res. 2022 Apr 13. PMID: 35418604.
2) Li et al. Alterations of gut microbiota diversity, composition and metabonomics in testosterone-induced benign prostatic hyperplasia rats. Mil Med Res. 2022 Mar 28;9(1):12. PMID: 35346378
3) Gu et al. High-Fat Diet Induced Gut Microbiota Alterations Associating With Ghrelin/Jak2/Stat3 Up-Regulation to Promote Benign Prostatic Hyperplasia Development. Front Cell Dev Biol. 2021 Jun 24;9:615928. PMID: 34249898
4) Magri et al. Multidisciplinary approach to prostatitis. Arch Ital Urol Androl. 2019 Jan 18;90(4):227-248. PMID: 30655633.
5) Hurst et al. Microbiomes of Urine and the Prostate Are Linked to Human Prostate Cancer Risk Groups. Eur Urol Oncol. 2022 Apr 18:S2588-9311(22)00056-6. PMID: 35450835.
6) Terrisse et al. Immune system and intestinal microbiota determine efficacy of androgen deprivation therapy against prostate cancer. J Immunother Cancer. 2022 Mar;10(3):e004191. PMID: 35296557
7) Porter CM et al. The microbiome in prostate inflammation and prostate cancer. Prostate Cancer Prostatic Dis. 2018 Sep;21(3):345-354. PMID: 29795140.
8) Shoemaker and Kim. Urobiome: An outlook on the metagenome of urological diseases. Investig Clin Urol. 2021 Nov;62(6):611-622. PMID: 34729961
A) Thank you for the suggestions, we were able to revise the manuscript with the papers you provided.
We also upload the final revised manuscript as a file.

Round 2
Reviewer 1 Report
Please include in the manuscript a statement concerning the lack of data concerning microbiome and IPSS and data comparing naive vs pharmacologically treated patients
Author Response
Please include in the manuscript a statement concerning the lack of data concerning microbiome and IPSS and data comparing naive vs pharmacologically treated patients
A) Thank you for comment. As your recommendation, we have revised and inserted the following sentences in the manuscript (The urinary microbiota and LUTS in male individuals)
[The urinary microbiota and LUTS in male individuals]
However, voided MSU cannot adequately characterize the male bladder microbiome, and the evidence of association between urinary microbiota and symptom severity (e.g., IPSS) is still lacking. Furthermore, there has been nor study including used naïve vs treated patients (alpha-blocker and other agents used in male LUTS), and neither evidence of clinical response versus clinical progression according to urinary microbiota after medication for male LUTS. Therefore, further research targeting specific microbes to identify their role in the development and treatment of LUTS is necessary.
Reviewer 2 Report
I appreciate the authors' efforts to improve the manuscript. The paper benefited from the revision. In my opinion, the manuscript now meets all the criteria for acceptance. Congratulations!
Author Response
Thanks with graceful comment.